

# The effect of warm summer 2012 on seasonal and annual methane dynamics in adjacent small lakes on the ice-free margin of Greenland

Sarah B. Cadieux[1,*], Jeffrey R. White[2] and Lisa M. Pratt[1]

[1]Department of Geological Sciences, Indiana University, Bloomington IN, USA

[2]School of Public and Environmental Affairs, Indiana University, Bloomington IN, USA

[*]now at: Earth and Environmental Science, University of Illinois at Chicago, Chicago Illinois, USA

*Correspondence to:* S. B. Cadieux (sbcadieux@gmail.com)

**Abstract**. In thermally stratified lakes, the greatest annual methane emissions typically occur during thermal overturn events. In July of 2012, Greenland experienced significant warming that resulted in substantial melting of the Greenland Ice Sheet and enhanced runoff events. This unusual climate phenomenon provided an opportunity to

examine the effects of short-term natural heating on lake thermal structure and methane dynamics and compare these observations with those from the following year when temperatures were normal. Here, we focus on methane concentrations within the water column of 5 adjacent small lakes on the ice-free margin of Southwest Greenland under open-water and ice-covered conditions from 2012-2014. Enhanced warming of the epilimnion in the lakes under open-water conditions in 2012 led to strong thermal stability and the development of anoxic hypolimnions in

each of the lakes. As a result, during open-water conditions, mean dissolved methane concentrations in the water column were significantly ($p < 0.0001$) greater in 2012 than in 2013. In all of the lakes, mean methane concentrations under ice-covered conditions were significantly ($p < 0.0001$) greater than under open-water conditions, suggesting spring overturn may be the period with the largest annual methane flux to the atmosphere. As the climate continues to warm, greater heat income and warming of lake surface waters will lead to increased

thermal stratification and hypolimnetic anoxia, which will result in increased water column inventories of methane. Additionally, continual warming will result in shorter ice cover durations, which may reduce the winter inventory of methane and lead to a decrease in total methane flux during ice-melt. These results suggest that inter-annual variation in ground-level air temperatures may be the primary driver of changes in methane dynamics because it controls both the strength of thermal stratification and duration of ice cover.

## 1 Introduction

Methane ($CH_4$) emissions from freshwater environments are expected to increase with warming climates (Juutinen et al., 2009; Yvon-Durocher et al., 2011; Yvon-Durocher et al., 2014; Tan and Zhuang 2015a,b) but quantitative modeled projections of emissions are poorly constrained (Bastviken et al., 2011; Rasilo et al., 2015; Sepulveda-





Jauregui et al., 2015). Observations of seasonal and annual lake $CH_4$ dynamics in the Arctic are necessary to define source estimates in models and understand the impact warming may have on greenhouse gas emissions. In the Arctic, small lakes (surface area < 10 km$^2$) are abundant (Downing et al., 2006; Downing, 2010) and emit substantially more $CH_4$ per unit area than larger lakes (Bastviken et al., 2004; Cole et al., 2007; Juutinen et al., 2009; Wik et al., 2016). On the ice-free margin of southwest Greenland, hundreds of thousands of Holocene lakes perched

on continuous permafrost cover the landscape (Anderson et al., 2001; Anderson and Stedmon, 2007; Jorgensen and Andreasen, 2007). As a result of amplified warming in the Arctic over the past 20 years (IPCC, 2013), Greenland has experienced significant mass loss of the Greenland Ice Sheet (Nghiem et al., 2012; van As et al., 2012; Hall et al., 2013; Hanna et al. 2013; Hanna et al., 2014). Despite the abundance of lakes on the ice-free margin of Greenland and intense changes to the landscape as the result of warming, there are only a few published studies that have

measured $CH_4$ in Greenlandic lakes (Walter Anthony et al., 2012; Webster et al., 2015; Cadieux et al., 2016; Goldman et al., 2016).

Microbial production of $CH_4$ by methanogens is dependent upon anoxia, temperature, and the amount and quality of organic carbon substrates (Liikanen et al., 2003; Duc et al., 2010; Borrel et al., 2011). A large proportion of the $CH_4$

produced in lakes is consumed by aerobic or anaerobic oxidation (Frenzel et al., 1990; Bastviken et al., 2002; Kankaala et al., 2007; Dzyuban, 2010; Martinez-Cruz et al., 2015). Aerobic microbial oxidation of methane (MOx) depends on the availability of both $CH_4$ and $O_2$, wherein higher MOx rates are usually found at the oxic/anoxic interface where both $CH_4$ and $O_2$ are present in high concentrations (Bastviken et al., 2002; Dzyuban 2010). Excess $CH_4$ that escapes MOx and reaches the upper mixed layer of the water column (epilimnion) is available for emission

to the atmosphere by molecular diffusion under open-water conditions. Emission by ebullition and plants results in a direct flux of $CH_4$ to the atmosphere with limited oxidation in the water column (Keppler et al., 2006; Walter et al., 2006; Walter et al., 2007; Nisbet et al., 2009; Wik et al., 2013).

In lakes, the amounts of $CH_4$ in the water column (hereafter referred to as inventory) and $CH_4$ available for diffusive

emissions (hereafter referred to as active $CH_4$) are strongly influenced by thermal stratification and seasonal overturns (Kankaala et al., 2007; López Bellido et al., 2009; Encinas Fernandez et al., 2014). Arctic lakes that are deep enough to stratify are usually dimictic (spring and autumn turnover) or cold monomictic (spring turnover) in areas without perennial ice cover. During thermal stratification, a lack of mixing between the epilimnion and anoxic hypolimnion suppresses gas transfer between these layers, allowing $CH_4$ to accumulate below the oxycline

(hereafter referred to as storage) (Fig. 1; Bastviken et al., 2004; Sepulveda-Jauregui et al., 2015). During mixing from autumn turnover, all $CH_4$ previously stored in the hypolimnion is susceptible to MOx and/or diffusion (Encinas Fernandez et al., 2014). Under ice cover, $CH_4$ can accumulate and is either stored under ice or within the ice (Fig. 1; Walter et al., 2006; Walter Anthony et al., 2012; Sepulveda-Jauregui et al., 2015). In spring, the break-up of the ice and mixing allows stored $CH_4$ to be oxidized or emitted from the system through diffusion or ebullition (Juutinen et

al., 2009; López Bellido et al., 2009; Jammet et al., 2015). Emissions of stored $CH_4$ during overturn events accounts



for up to 40% of the total annual flux in lakes globally (Michmerhuizen et al., 1996; Juutinen et al., 2009; López Bellido et al., 2009; Encinas Fernandez et al., 2014; Jammet et al., 2015).

Climate changes will result in variations in heat balance, temperature profiles and vertical mixing in lakes (Jankowski et al., 2006; MacIntyre et al., 2009; Hinkel et al., 2012; Butcher et al., 2015), causing many variations to both lake structure (Livingstone 2003; Coats et al., 2006) and $CH_4$ dynamics. Over the last three decades, increasing atmospheric temperatures have resulted in increased lake temperatures and decreases in ice cover (Weyhenmeyer et al., 2011; Kraemer et al., 2015). Warming of surface waters will lead to increased thermal stratification and hypolimnetic anoxia, which should cause higher inventories of dissolved $CH_4$ stored in the hypolimnion of the water column. In addition to changes to thermal structure, warming and thawing of permafrost in the Arctic may allow organic carbon, nutrients and ions that were previously frozen in sediments to be transported into aquatic systems and become available for microbial utilization (Kokelj et al., 2009; Lougheed et al., 2011; Weyhenmeyer et al., 2011). Experimental laboratory incubation studies have also demonstrated that microbial $CH_4$ production significantly increases with increased temperature (Duc et al., 2010; Lofton et al., 2013; Fuchs et al., 2016).

The anomalously warm summer in Greenland 2012, which resulted in substantial surface melt of the Greenland Ice Sheet (GIS) (Nghiem et al., 2012; Hanna et al., 2014) provides an opportunity to examine the effects of surface water heating on $CH_4$ dynamics in lakes. In this study, we quantify the depth inventories of $CH_4$ under both open-water and ice-covered conditions for 5 adjacent small lakes on the ice-free margin of Southwest Greenland from 2012 to 2014. In doing so, we are able to look at differences in $CH_4$ spatially among the lakes, seasonally by comparing open-water conditions to ice-covered conditions, and annually. We hypothesized that warmer conditions in the summer of 2012 would lead to significantly greater $CH_4$ concentrations under open-water conditions than in 2013. This work provides the first measurements of dissolved $CH_4$ concentrations under both open-water and ice-covered conditions for consecutive years in small, Arctic lakes.

## 2 Study Area

The study area lies between the village of Kangerlussuaq, Greenland at the head of Søndre Strømfjord and the active terminus of the Russell Glacier (Fig. 2). The region has continuous permafrost extending from 50 cm below the surface to 130 m at Kangerlussuaq and 500 m at the ice sheet (Jorgensen and Andreasen, 2007). Precipitation in the Kangerlussuaq region is low, with annual precipitation < 150 mm $yr^{-1}$. Dwarf shrubs (*Salix*, *Vaccinium* and *Betula*) and graminoids (*Carex* and *Calamagrostis*) dominate the tundra vegetation in the region.

This paper focuses on 5 small lakes (surface area < 3 ha, maximum depth < 8 m), with a range of different morphometries and aquatic chemistries (Table 1). The lake names used herein are informal. The study lakes are part of a series of lakes within a narrow valley overlying a structural shear zone extending from the Russell Glacier to the Søndre Strømfjord (Fig. 2). The lakes are, at most, 6 km apart and are subject to the same climatic forcing. All lakes in the study are dimictic, exhibiting ice cover from late October to early June. Currently the study lakes are all



hydrologically closed basins, with no active inflow or outflow channels observed from 2011-2014. Groundwater seepage into the lakes is assumed to be limited due to continuous permafrost.

110

In summer 2012, a blocking high-pressure system formed a heat dome over Greenland, leading to widespread surface melting of the GIS (Hall et al., 2013; Hanna et al., 2014; Nghiem et al., 2012). As a result, this circulation pattern produced mean daily ground-level air temperatures in Kangerlussuaq that were the highest on record (Hanna et al., 2014). Weather data from a local Geological Survey of Denmark and Greenland (GEUS) station (van As et al., 2011; Fig. 2) showed that mean monthly ground-level temperatures for June-August in 2012 were ~2° C higher than in 2013 (Fig. 3). Correspondingly, mean air pressure from May to August of 2012 was higher than in 2013 (Fig. 3). Mean monthly wind speeds in the Kangerlussuaq region ranged from 1.52 m s$^{-1}$ in June 2012 to 3.51 m s$^{-1}$ in May of 2013 (Fig. 3). At the Kangerlussuaq airport, average temperatures from May to August of 2013 were within the range observed from 1996-2011, suggesting 2013 weather is typical for the region (data retrieved from weatherunderground.com).

## 3 Methods

### 3.1 Sample collection and water chemistry analysis

Lakes were sampled in July of 2012 and 2013 and in April of 2013 and 2014 in order to measure both summer and winter stratification. Hereafter, July samplings will be referred to as open water and April samplings as ice covered. Only EVV Upper lake and Potentilla lake were sampled during all four sampling dates. All samples and measurements were taken at a location above maximum water depth ($Z_{max}$). Under open-water conditions, samples and measurements were collected using an inflatable Alpaca raft (Anchorage, AK, USA), and under ice-covered conditions, when each lake was covered by ~ 2 m of ice, a hole (~30 cm in diameter) was augered through the ice in order to sample. Temperature (T, °C), pH, dissolved oxygen (DO, mg L$^{-1}$), oxidation-reduction potential (ORP, mV) and specific conductivity (mS cm$^{-1}$) were measured using a YSI 6093 Data Sonde (Yellow Springs Inc., Yellow Springs, OH, USA) deployed at vertical intervals of 0.5 m depth.

Water for chemical analysis was collected using an electronic submersible pump. Samples were frozen, and transported to Indiana University, where all chemical analyses were conducted. Dissolved organic carbon (DOC) was analyzed from filtered samples that were acidified using hydrochloric acid (HCl), and analyzed via high-temperature oxidation using a Shimadzu total organic carbon analyzer (corresponding Method Detection Limit (MDL) = 0.15 mg C L$^{-1}$). Concentrations of ions were analyzed using a Dionex ICS 2000 Ion Chromatograph using a CS12A analytical cation column, CSRS 300 4 mm suppressor, and 20 mM methanesulfonic acid eluent for cations and AS11-HC analytical anion column, ASRS 4 mm suppressor and 30 mM potassium hydroxide eluent for anions.





### 4.2 CH$_4$ collection

With the exception of Potentilla lake under ice-covered conditions in 2014, water samples for dissolved CH$_4$ in the water column were collected using an electronic submersible pump. Samples were collected over multiple depth intervals within the water column and were immediately stripped in the field using a headspace-equilibrium technique (Westendorp 1985) to extract CH$_4$ from water using a 1 L Erlenmeyer flask. Headspace gas in the flask was displaced into a Cali-5-Bond bag using surficial lake water (Cadieux et al., 2016). Under ice-covered conditions

in 2014, dissolved CH$_4$ in Potentilla lake was collected using a string of passive diffusion bags (PDBs) deployed in the lake for 5 days in order to obtain a high-resolution profile of dissolved CH$_4$ in the water column (Cadieux et al., 2016; Goldman et al., 2016). The PDBs are composed of a polyethylene membrane with a protective plastic mesh and are commercially available from EON Products Inc. (Georgia, USA). Further details regarding PDB methodology, preparation and applicability can be found in Goldman et al. (2016).


Profundal sediment samples were taken from Z$_{max}$ of each lake during ice-covered conditions and open water conditions in 2013 using a Wildco push-coring device. Cores were transferred back to the laboratory at Kangerlussuaq International Science Support (KISS) facility and immediately refrigerated at 4° C and processed within 24 hours of collection. Dissolved gas in the sediment was sampled using an equilibrium gas stripping method

similar to that used for the water-column CH$_4$. Sediment cores were sub-sectioned into 6-10 cm intervals and each subsection was put into a 8 L Nalgene bottle with zero-CH$_4$ water to create 2 L sediment-water slurry, which was vigorously hand shaken for 5 minutes to displace gas from the sediment-slurry into the headspace (Cadieux et al., 2016). The volume of pore water in the sediment core was calculated by drying an additional subsection of sediment. The concentration of CH$_4$ in the sediment cores was calculated as moles of CH$_4$ per unit volume of pore

water in the sediment.

Littoral sediment CH$_4$ bubble samples were collected during open-water conditions of both 2012 and 2013 by physically disturbing the sediment in order to release entrained gas bubbles. Gas bubbles were collected using a large plastic funnel (28 cm diameter) with a gas-tight sampling tube and 3-way Luer-Lok valve attached to the neck

(Cadieux et al., 2016). We were unable to quantify the volume of sediment samples, therefore concentrations of CH$_4$ in gas collected from littoral sediments cannot be converted into pool size of CH$_4$ in the littoral sediments.

### 4.3 Concentration of CH$_4$

The concentrations of water-column CH$_4$ and sediment CH$_4$ were measured using a Los Gatos Research (LGR)

Methane Carbon Isotope Analyzer (MCIA) (LGR, Mountain View CA, USA) that was operated at KISS (Cadieux et al., 2016). All samples were processed within 24 hours of collection. The total concentration of CH$_4$ in each sample was corrected for dilution and calculated from the sum of the measured headspace partial pressure and the dissolved CH$_4$ remaining after gas stripping, according to Henry's law using values from Lide and Fredrikse (1995). Instrumental uncertainty on CH$_4$ concentrations from the MCIA was ± 0.5 ppmv, which is one standard deviation of

the values for gas standards analyzed during sample runs.



### 4.4 Inventory of dissolved CH$_4$

Bathymetric data were collected under open-water conditions in 2013 using a LOWRANCE HDS-5 Gen2 depth-sonar built with GPS and processed by ciBio Base software by Contour Innovations LLC (Minneapolis, MN, U. S.). The area and volume of water were derived and measured from bathymetric data. The total inventory of dissolved CH$_4$ in each lake was calculated by multiplying CH$_4$ concentrations for each depth interval by the volume of each depth interval. It was assumed that CH$_4$ concentrations within each depth interval were homogenous both horizontally and vertically. During open-water conditions, the depth intervals for active and storage pools were defined by redox conditions, where the storage depths are defined as intervals with DO < 1 mg L$^{-1}$. The active depths were associated with the oxic epilimnion where dissolved gases are susceptible to diffusive exchange with the atmosphere and exposed to atmospheric oxygen. Under ice cover, the size of the CH$_4$ storage pool was assumed to be that contained in the volume of water below ice.

### 4.5 Statistical Analyses

Statistical analyses were made using IBM SPSS Statistics. Concentrations of CH$_4$ and chemical variables for all study lakes during each season were assessed for normal distribution via the Kolmogorov-Smirnov test, and were found to be non-normally distributed. Student's t test of unequal variance was used for testing statistically significant differences in concentrations of CH$_4$, temperature and DO between seasons and years. Systematic changes in aquatic chemistry and CH$_4$ concentrations were analyzed using linear regression.

## 5 Results

### 5.1 Thermal structure and DO profiles

Under open-water conditions, all lakes were thermally stratified in 2012 (Fig. 4). Thermal stratification, expressed in terms of temperature difference ($\Delta T = T_{0m} - T_{zmax}$), was > 12 °C in all lakes except for EVV Lower lake, where $\Delta T \approx$ 7 °C. For all lakes under open-water conditions, epilimnetic temperatures were significantly warmer ($n=30$, $p <$ 0.0001) and bottom waters were cooler in 2012 than in 2013 (Fig. 4). Under open-water conditions in 2013, various levels of thermal stratification were observed, ranging from thermally stratified in EVV Upper lake and Potentilla lake ($\Delta T \approx 9$°C) to isothermal in EVV Lower lake ($\Delta T \approx 1$ °C; Fig. 4). Under ice-covered conditions in both 2013 and 2014, lakes were nearly isothermal ($\Delta T < 2$ °C) except for Potentilla lake which was weakly thermally stratified under ice cover, with $\Delta T \approx 4$ °C in both years.

Clinograde DO profiles were observed in all lakes under open-water conditions in 2012, wherein DO was saturated and in equilibrium with the atmosphere in the surface waters and became increasingly undersaturated down the water column (Fig. 5). Anoxia (DO < 0.5 mg L$^{-1}$) was measured in the bottom waters of all lakes except for Teardrop lake. In EVV Upper lake, Potentilla lake and South Twin lake, the bottom 2 to 2.5 m of the water column was anoxic. Similar clinograde DO trends were observed under open-water conditions in 2013, although stratification was weaker, (Fig. 5) and anoxia was limited to the bottom 1 m of the water column in EVV Upper lake





and Potentilla lake. All lakes exhibited complete anoxia below the ice with the exception of Potentilla lake in both years. Measurable DO was observed under ice in Potentilla lake, with a clinograde profile from suboxic conditions (7.0 – 5.0 mg L$^{-1}$) below the ice to 4.5 m and anoxic conditions from 5.5 m to the sediment/water interface.


## 5.2 Aquatic chemistry

Ionic composition varied markedly lake to lake as well as seasonally and annually (Table 2). According to a salinity classification scheme based on specific conductivity (Stewart and Kantrud, 1971), under open-water conditions in 2012, two of the study lakes were dilute (.04-0.5 mS cm$^{-1}$; EVV Upper and Potentilla), one was slightly brackish (0.5-2 mS cm$^{-1}$; EVV Lower), and two were moderately brackish salinity, with maximum specific conductivity exceeding 2 mS cm$^{-1}$ (Table 2). The anion abundance followed $HCO_3^- > DOC^-/Cl^- > SO_4^{2-}$ in most of lakes, regardless of conductivity. $DOC^-$ represents the estimated charge on DOC based upon the anionic charge deficit (Driscoll and Newton, 1985) (Fig. 6). EVV Upper lake was the only lake where $SO_4^{2-} > Cl^-$. Sulfate accounted for 12 % of the total anion abundance in EVV Upper lake, relative to < 3 % in the other lakes. Cation abundance in dilute lakes followed $Ca^{2+} > Mg^{2+} > Na^+ > K^+$, whereas in slightly brackish lakes, $Mg^{2+} > Na^+ > Ca^{2+} > K^+$ was observed (Fig. 6). Overall, conductivity and ionic compositions were higher under ice-covered conditions than open-water conditions (Table 2). However, under open-water conditions in 2012, mean specific conductivity was significantly higher in all lakes than in 2013 ($n$=53, $p < 0.0001$).

Under open-water conditions in 2012, concentrations of DOC ranged from median of 11 mg L$^{-1}$ (Potentilla lake) to 92 mg L$^{-1}$ (Teardrop lake; Table 2). Under ice-covered conditions, DOC was higher than open-water conditions (Table 2). No consistent trends were observed for DOC between open-water conditions in 2012 and 2013.

## 5.3 Concentrations of CH$_4$

### 5.3.1 Dissolved water column CH$_4$

Concentrations of dissolved CH$_4$ were significantly greater under open-water conditions in 2012 than in 2013 ($n$=38, p=0.008; Fig. 6). Under open-water conditions in 2012, dissolved CH$_4$ concentrations in the surface waters of the study lakes ranged from 1.2 – 28 μM (Fig. 7). In all of the lakes, CH$_4$ concentrations increased down the water column and were greatest in the bottom waters (Fig. 7). The highest concentration of dissolved CH$_4$ (640 μM) was observed at 4.5 m in South Twin lake. Similar to 2012, in 2013 CH$_4$ concentrations increased down the water column in all lakes except for Teardrop lake. Under open-water conditions in 2013, concentrations of CH$_4$ in at the water-air interface ranged from 0.88 – 3.5 μM (Fig. 7). In Teardrop lake, CH$_4$ was < 10 μM though the water column, ranging from 1.4 to 8.1 μM (Fig. 7). The highest concentration of CH$_4$ (150 μM) was observed at 5.0 m in EVV Upper lake.


Under ice-covered conditions, the mean concentrations of dissolved CH$_4$ were significantly greater than under open-water conditions ($n$=29, $p$ <0.0001; Fig. 7). In 2013, CH$_4$ concentrations under ice cover were relatively uniform



down the water column in EVV Upper lake (102-150 μM) and EVV Lower lake (340-360 μM) and ranged from

450-690 μM in South Twin lake. Only in Potentilla lake did $CH_4$ increase down the water column from 1.3 μM

under the ice to 812 μM at the sediment-water interface (Fig. 7). The mean concentrations of dissolved $CH_4$ under

ice cover in 2013 were significantly greater than in 2014 (n=29; $p < 0.0001$; Fig. 7). In 2014, dissolved $CH_4$

concentrations increased down the water column from below ice cover to the sediment-water interface in all of the

three lakes, from 49 – 68 μM in EVV Upper lake, 190-360 μM in Teardrop lake and 0.3-220 μM in Potentilla lake

(Fig. 7).


Linear relationships between aquatic chemistry and $CH_4$ concentrations were weak (Fig. 8; $r^2$ values for regressions

ranged from 0.20 – 0.35). The highest $CH_4$ concentrations were correlated to both lower temperatures and lower

DO, however the lowest $CH_4$ concentrations did not correlate to highest temperatures or DO (Fig. 8a & b). Overall,

high concentrations of $CH_4$ were related to high conductivity and DOC (Fig. 8c & d).


### 5.3.2 Whole-lake inventories and pool sizes of $CH_4$

In all the lakes, the total inventory of $CH_4$ in the water column was higher under ice-covered conditions in 2013 and

lower under open-water conditions in 2013 (Fig. 9 a). Under open water conditions, total dissolved $CH_4$ was 2 – 20

times greater in 2012 than in 2013. During all sampling dates, EVV Upper lake had the lowest $CH_4$ inventory among

the lakes (Fig. 9 a).

Under open water conditions, the inventories of both the stored and active pools of $CH_4$ varied between lakes and

years (Fig. 9 b & c). In 2012, the majority (> 50%) of the total dissolved $CH_4$ inventory in the water column of EVV

Upper lake and Potentilla lake was associated with the hypolimnion (Fig. 9 b). EVV Lower lake was the only lake in

2012 wherein the majority of the total inventory of $CH_4$ occurred in the active pool of the epilimnion. In 2013, with

the exception of EVV Upper lake, the majority of the total $CH_4$ inventory was in the active pool in all of the lakes

(Fig. 9 c). EVV Upper lake was the only lake where the majority of the total $CH_4$ inventory was associated with the

hypolimnion (Fig. 9 c). We saw no evidence of holes or moats in the ice in any of the lakes at the end of ice-covered

conditions in 2013 or 2014, suggesting that the total inventories are likely relatively well conserved.


### 5.3.3 Sediment $CH_4$

In all lakes, the concentrations of $CH_4$ in porewaters of profundal sediments were an order of magnitude greater than

dissolved $CH_4$ concentrations in the water column (Table 3). Under ice cover in 2013, profundal $CH_4$ concentrations

were greater than under open-water conditions in 2013 for all of the lakes. Littoral sediment $CH_4$ gas bubble

concentrations in Potentilla and South Twin lake were greater in 2012 than 2013, whereas in Teardrop lake,

concentrations of $CH_4$ were greater in 2013 (Table 3). Overall, the maximum littoral sediment $CH_4$ gas bubble

concentration was 473,000 ppmv in EVV Upper lake in 2013 and the minimum was 166,000 ppmv in Potentilla

2013.





## 6 Discussion

### 6.1 Spatial variation in aquatic chemistry and methane concentrations

Aquatic chemistry of the study lakes during mid-summer varied considerably among lakes with no discernable spatial trends. Chemical characteristics in each lake likely reflect interactions between basin-specific factors such as bedrock geology, basin morphometry and macrophyte community composition. Geology in the Kangerlussuaq region has been generally described as dominated by granodioritic gneisses (Anderson et al. 2001; Jensen et al. 2002). Despite large differences in conductivity, $HCO_3^-$ was the dominant anion in the lakes. However, we have observed and measured sections of $SO_4^{2-}$ minerals locally, occurring in orange-brown, thinly bedded outcrops interpreted as weathered, sulfide-rich metasediments (unpublished data). Elevated $SO_4^{2-}$ in only EVV Upper lake leads us to suspect there is sulfide-rich, metasedimentary bedrock in the basin of this lake that has contributed to the anomalously high $SO_4^{2-}$. In addition, pods of marble have been described in the region previously (Taylor and Kalsbeek 1990), although they were not identified locally. These localized marble pods could be responsible for increased $Ca^{2+}$ in Teardrop lake. Previous work in the Kangerlussuaq regions suggests that anomalously high concentrations of DOC may be associated with abundance of littoral macrophytes (Lim and Douglas 2003; Lim et al 2005; Anderson and Stedmon 2007). The lowest DOC concentrations were from Potentilla lake, which had a comparatively lower density of macrophytes in comparison to the other lakes. The relative density of macrophyte communities in these hydrologically closed basins may strongly influence DOC concentrations in the lakes.

In the water column, $CH_4$ concentrations are related to both conductivity and DOC, wherein overall, higher $CH_4$ concentrations are related to higher conductivity and DOC (Fig. 7). However, we do not find a direct relationship between maximum $CH_4$ concentration and aquatic chemistry variables. Under open-water conditions of 2012, the maximum $CH_4$ concentration in the water column was from the hypolimnion of South Twin lake. While South Twin lake did exhibit relatively high conductivity, DOC concentrations were relatively low. Similarly, the highest $CH_4$ concentration was measured in Potentilla lake in 2013 under ice-covered conditions, corresponding to the lowest conductivity and DOC measured in the study. The lowest concentrations of $CH_4$ in all of the lakes were from epilimnetic waters, despite the wide range in aquatic chemistry lake-to-lake. In addition, EVV Upper lake had significantly higher concentrations of $SO_4^{2-}$ than the other lakes in the study. Competition for substrates favors sulfur reduction (SR) and methanogenesis typically does not occur until $SO_4^{2-}$ is exhausted and SR rates have decreased (Lovely & Klung 1983, Lovely & Klung 1986, Scholten et al., 2002, Ward & Winfrey 1985). However, EVV Upper lake did not have the lowest concentrations of $CH_4$ in the water column, suggesting there was sufficient reduced carbon substrates to fuel both SR and methanogenesis. Therefore, while aquatic chemistry in the water column may be a factor influencing $CH_4$ production, it alone is insufficient to explain the variation in $CH_4$ concentrations observed lake-to-lake, as well as seasonally and annually.

### 6.2 Effect of temperature on lake stratification

Ground-level air temperatures strongly influence the thermal stratification of lakes during the open-water season. Warm ground-level air temperatures during open-water conditions in 2012 (Fig. 3) resulted in epilimnetic




temperatures being 1.5 – 5 °C higher in all of the study lakes relative to open-water conditions in 2013 (Fig. 4). Increased epilimnetic temperatures under open-water conditions in 2012 are consistent with both predictive models and measured temperatures indicating that warming climates result in higher epilimnetic temperatures (Honzo and Stefan, 1993; Fang et al., 2009; Jankowski et al., 2006; Adrian et al., 2009; Coats et al., 2006). As a result of warmer

epilimnetic waters, stronger thermal stratification occurred during open-water conditions in 2012 than in 2013, with $\Delta T$ 4 – 9 °C higher in 2012 (Fig. 4).  In addition, wind speeds were significantly lower and air pressures were higher during open-water conditions in 2012 compared with 2013 (Fig. 3), leading to reduced mixing of the water column and greater heat transfer to shallower epilimnia. Temperature and thermal structure strongly influence DO concentrations in lakes, wherein stronger thermal stratification leads to increased anoxia in the hypolimnion (Hanson

et al., 2006; Adrian et al., 2009; Foley et al., 2012). For example, the extremely warm European summer of 2003 resulted in stronger thermal stratification and hypolimnetic DO depletion in Swiss lakes (Jankowski et al., 2006). Similarly, as a result of strong thermal stratification in 2012, the hypolimnia of EVV Upper lake, EVV Lower lake, Potentilla and South Twin lake were anoxic. For comparison, 1-34 % of water column in the lakes was anoxic in 2012, whereas the percentages decreased to 0-7 % in 2013.


### 6.3 Effects of temperature on CH$_4$

Ground-level air temperature differences result in warmer surface waters and stratification differences between 2012 and 2013 but a weak linear relationship is observed between water temperatures and dissolved CH$_4$ concentrations (Fig. 3-7). Both the highest and lowest CH$_4$ concentrations are observed in waters < 5 °C. In freshwater

environments, the concentration of dissolved CH$_4$ reflects the balance between CH$_4$ production and CH$_4$ consumption by anaerobic or aerobic oxidation (Duc et al., 2010; Dzyuban, 2010; Encinas Fernandez et al., 2014; Kankaala et al., 2007; Martinez-Cruz et al., 2015; Segarra et al., 2015; Smemo and Yavitt 2011). Methane production is affected by temperature, where higher temperatures result in increased production (Duc et al., 2010). However, methanogenesis only occurs under anaerobic conditions (Borrel et al., 2011; Valentine et al., 1994; Yvon-

Durocher et al., 2011). Under open-water conditions in all of the lakes, the majority of the water column is oxygenated (Fig. 3-2), therefore production was likely minimal in the water column in both 2012 and 2013. Concentration of CH$_4$ and DO are linearly related, wherein highest concentrations occur in anoxic waters and decrease with increasing DO (Fig. 7), suggesting MOx driving the concentration of CH$_4$ in the water column. However, consumption of CH$_4$ by microbial methane oxidation is not strongly influenced by temperature (Duc et al.,

2010) and has only been demonstrated to increase with increasing temperatures under CH$_4$ saturated conditions (Lofton et al., 2013).

Despite the absence of a strong linear relationship between water temperature and CH$_4$ concentrations, warmer ground-level air temperatures correspond with increased CH$_4$ both in the water column and the sediments. Under

open-water conditions, CH$_4$ concentrations in the water column were significantly greater in 2012 than in 2013 (Fig. 8), corresponding with increased ground-level air temperatures. Similarly, under ice-covered conditions, ground-level air temperatures were ~ 6 °C higher in 2013 than in 2014 and CH$_4$ concentrations in the water column were



greater for the 2 lakes in which there is data for both years (Fig. 9). The $CH_4$ concentration differences occur throughout the water column, but are more pronounced in the bottom waters close to the sediment-water interface. It

is possible that increased ground-level air temperatures result in increased production of $CH_4$ in the profundal sediments, which lead to increased concentrations in the bottom waters of each lake. However, under open-water conditions in 2012, the bottom water temperatures were colder than in 2013, suggesting profundal sediments were not warmer due to increased ground-level air temperatures. More likely, the higher $CH_4$ concentrations in the bottom waters during open-water conditions in 2012 were the result of increased thermal stratification and subsequent

anoxia, allowing a buildup of $CH_4$ in the bottom waters.

The concentrations of $CH_4$ from profundal sediments during ice-covered conditions in 2013 were greater than from open-water conditions in 2013 (Table 3). While ground-level air temperatures were significantly colder during the 2013 ice-covered season compared to the 2013 open-water season (Fig. 3), the higher concentrations of $CH_4$ during

ice-covered conditions may be a relic of the anomalously warm conditions from the previous open-water conditions in 2012. In the littoral sediments, where there is data for consecutive years, gas bubble $CH_4$ concentrations from 2012 were higher than in 2013 (Table 3), further suggesting that warmer ground-level air temperatures result in increased $CH_4$ production, consistent with experimental studies of methanogenesis response to higher temperature (Duc et al., 2010; Hoj et al., 2008; Lofton et al., 2013).


Without temperature data for profundal and littoral sediments, it is impossible to directly determine whether warmer temperatures result in an increase in $CH_4$ production or if other factors may influence production of $CH_4$. However, because bottom water temperatures were colder during open-water conditions in 2012 than in 2013 (Fig. 4), it is unlikely that profundal sediments were warmer in response to warmer ground-level air temperatures. In addition to

temperature, methanogenesis is also influenced by the amount and quality of organic carbon substrates (Borrel et al., 2011; West et al., 2012). Lakes in this study are all embedded within continuous permafrost, with an active layer < 0.5 m thick. It is possible that the anomalously warm conditions in 2012 resulted in warming and thickening of the active permafrost layer, which could have caused organic carbon, nutrients and ions to enter the lakes and be available for microbial utilization (Adrian et al., 2009; Kokelj et al., 2009; Lougheed et al., 2011; Weyhenmeyer and

Karlsson, 2009). However, under open-water conditions, DOC was only higher in one lake in 2012 compared to 2013 and DOC concentrations during ice-covered conditions in 2013 and 2014 were similar (Table 2). Significantly higher specific conductivity during open-water conditions in 2012 compared to 2013 was observed for all of the lakes (Table 2), which could be attributed to an additional source from permafrost thaw. Increases in DOC and conductivity were observed in thaw ponds in Western Siberia during the anomalous hot summer of 2012, but were

attributed to evapoconcentration effects (Pokrovsky et al., 2013). In the Greenlandic lakes, significant water level changes between the two consecutive years of this study were not observed, so it is unlikely that the higher conductivity in 2012 was the result of evaporation.





### 6.4 Effects of stratification on CH$_4$

Enhanced thermal stratification and anoxia during open-water conditions in 2012 resulted in significantly higher CH$_4$ concentrations in the water column. The most notable difference in CH$_4$ concentrations between 2012 and 2013 occurred within the bottom waters, which were anoxic in 2012. Under open-water conditions in 2012, the amount of CH$_4$ stored in the hypolimnion was 2 to 300 times higher than in 2013. The higher CH$_4$ concentrations may be the result of increased CH$_4$ production due to more extensive anaerobic conditions. However, the higher CH$_4$ concentrations were more likely the result of larger pools of anoxic waters in 2012 allowing for a buildup of CH$_4$ that resulted in increased CH$_4$ storage in the water column (Fig. 8). During fall overturn, the storage pool of CH$_4$ is susceptible to diffusion and/or oxidation (Encinas Fernandez et al., 2014; Kankaala et al., 2007; López Bellido et al., 2009). If half of the stored CH$_4$ is emitted during fall overturn, as suggested by results from Fernandez et al., (2014), the autumn overturn CH$_4$ flux would be significantly larger in 2012 than that in 2013 as a result of the more extensive anoxia in 2012.

Weaker thermal stratification during open water conditions in 2013 meant that 93 –100 % of the water column in the lakes had DO concentrations > 1 mg L$^{-1}$. When CH$_4$ diffusing from anoxic sediments reaches oxic sediment or water, the majority of it is oxidized (Bastviken et al., 2002; Dzyuban, 2010; Kankaala et al., 2007). MOx is highly efficient at consuming CH$_4$ thereby lowering CH$_4$ concentrations. The percentage of CH$_4$ oxidized can be estimated by assuming that CH$_4$ at the water-air interface originates from diffusion through the water column from the profundal sediments. Given that an increased proportion of the water column was oxic under open-water conditions 2013, we initially hypothesized that the percentage of CH$_4$ oxidized was greater in 2013 than in 2012. However, under open-water conditions in 2013, the percentage of CH$_4$ oxidized was similar to, or less, than in 2012. Consistent with CH$_4$ oxidation rates from Alaskan lakes, MOx controls CH$_4$ concentrations when DO is present (Martinez-Cruz et al., 2015). Under warm conditions in 2012, not only were CH$_4$ concentrations in sediments and anoxic waters elevated, but MOx was also higher. Several studies suggest that MOx is important for mitigating CH$_4$ emissions to the atmosphere (Martinez-Cruz et al., 2015; Milucka et al., 2015; Segarra et al., 2015). However, despite the likelihood that MOx was efficient in 2012 under warmer conditions, CH$_4$ concentrations were higher in 2012 than in 2013.

While significant variations in CH$_4$ concentrations and inventories were observed under between the consecutive years under open-water conditions, the amount of CH$_4$ stored under ice cover was significantly greater than that stored in the anoxic hypolimnion under open-water conditions (Fig. 3-6). For South Twin lake, the CH$_4$ storage under ice cover in 2013 was more than an order of magnitude greater than that stored in the hypolimnion during open-water conditions in 2012, and in Lower lake it was two orders of magnitude greater (Fig. 3-6). In these small Greenlandic lakes, emissions during spring overturn reflect the largest potential flux of CH$_4$ to the atmosphere. Similarly, in other Arctic lakes, CH$_4$ emissions during spring overturn after ice-breakup are usually larger than CH$_4$ emissions during autumn overturns (Juutinen et al., 2009; López Bellido et al., 2009).





### 6.5 Implications for a warmer Arctic

In the Arctic, lakes are ice covered for more than 8 months of the year (Belzile et al., 2001). The study lakes here are ice covered ~10 months of the year. At sampling under ice-covered conditions, lakes have been covered ~8 months.

In each of the lakes, $CH_4$ concentrations are significantly higher under ice-cover conditions compared to open-water conditions (Fig. 3-4), which is also observed in other Northern latitude lakes that are ice covered the majority of the year (Juutinen et al., 2009; Martinez-Cruz et al., 2015). Ice cover impedes gas exchange between the water and the atmosphere, promoting buildup of $CH_4$ in the water column leading to increased $CH_4$ storage (Bastviken et al., 2004; Juutinen et al., 2009; Martinez-Cruz et al., 2015; Phelps et al., 1998). No holes or moats were observed in the ice

cover during sampling, therefore the total inventory of $CH_4$ in the water column under ice-covered conditions was stored. Similar to stored $CH_4$ during stratification in open-water conditions, $CH_4$ stored under ice is susceptible to emission to the atmosphere during spring overturns during and after ice breakup.

Projected climate change is expected to change ice cover characteristics in lakes (Fang and Stefan 2009; Mueller et

al., 2009). Ice coverage duration has already decreased for many lakes as ground-level air temperatures have increased (Bertilsson et al., 2013; Weyhenmeyer et al., 2011). As perennially ice covered lakes begin to develop open water periods for at least some portions of the year, the number of seasonally frozen lakes will increase (Mueller et al., 2009). Our results suggest that changes in the duration of seasonal ice cover will, in turn, result in changes in inventories of under-ice $CH_4$. As the duration of ice cover decreases, the amount of $CH_4$ stored under ice

cover will likely decrease due to the shorter time for accumulation, potentially reducing the amounts of $CH_4$ emitted during ice-breakup and spring overturn.

In addition to a decrease in ice cover, our results also suggest an increase in ground-level air temperatures will result in enhanced thermal stability and anoxia in Arctic lakes, as we observed during open-water conditions in 2012. The

duration of open-water thermal stratification will also likely increase in concert with the decrease in ice cover. The combined effect is likely to be higher $CH_4$ inventories in the water column during open-water conditions in Arctic lakes. Currently, small lakes emit substantially more $CH_4$ per unit area than larger lakes during open-water conditions (Bastviken et al., 2004; Cole et al., 2007; Juutinen et al., 2009). Small, shallow likes are more susceptible to thermal change due to increased ground-level air temperatures and will likely continue to be major $CH_4$

contributors to the atmosphere. Our results suggest that increased warming in the Arctic will result in larger emissions of $CH_4$ to the atmosphere during autumn overturn.

### 7 Conclusions

Over the past half century, the Arctic has warmed at a rate greater than the global average, and climate models

predict further polar amplification, with the Arctic continuing to warm at a faster rate than other regions. The anomalously warm summer of 2012 in Greenland resulted in significantly higher $CH_4$ concentrations under open-water conditions in a series of small lakes compared to the following year. Stronger thermal stratification under warmer conditions lead to increased $CH_4$ storage in the lakes. With impending warming climate, increased





stratification and CH$_4$ storage in lakes will likely lead to greater potential fluxes during fall overturn. However, in these small, seasonally ice covered Arctic lakes, the greatest concentrations of CH$_4$ in the water column are occurring under ice-covered conditions. Changes in seasonal ice cover will result in changes in under ice CH$_4$ inventories, and consequently lead to variations in the amount of CH$_4$ emitted during ice-breakup and spring overturn. These results suggest that inter-annual variation in ground-level air temperatures may be the primary driver of changes in methane dynamics because it controls the both the strength of thermal stratification and duration

of ice cover.

**8 Acknowledgements**

Funding for this work was provided by NASA ASTEP Grant #NNX11AJ01G. We thank Contour Innovations LLC for assistance with bathymetric maps and Amy Goldman, Seth Young and Yongbo Peng for assistance in the field

with sample collection. For logistical support, we thank Polar Field Services, Inc., Kangerlussuaq International Science Support and Ruth Droppo for logistical support. Weather data from the Programme for Monitoring of the Greenland Ice Sheet (PROMICE) and the Greenland Analogue Project (GAP) was provided by the Geological Survey of Denmark and Greenland (GEUS) at http://www.promice.dk.

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

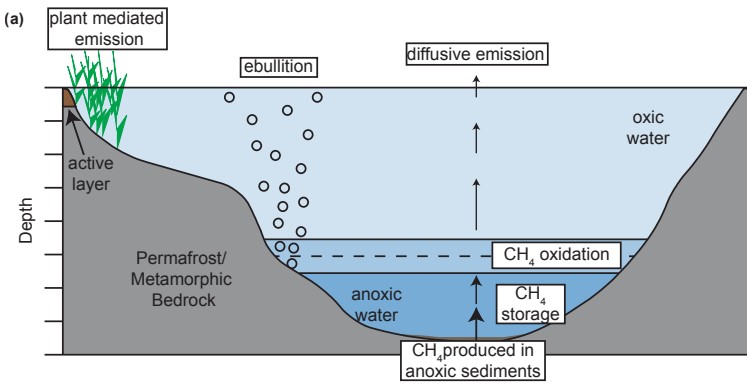

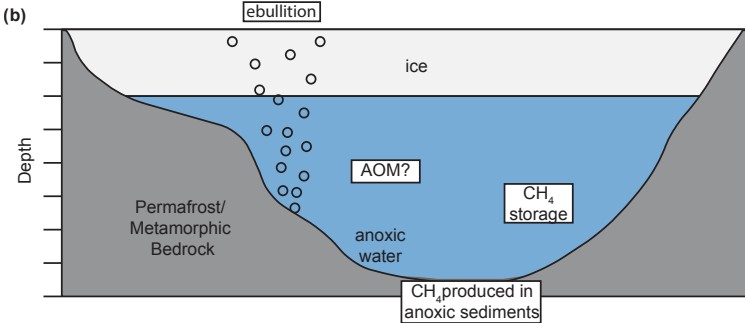

Figure 1. Methane emission pathways and dynamics in a bedrock controlled thermokarst lacustrine system under both open-water and ice-covered conditions.



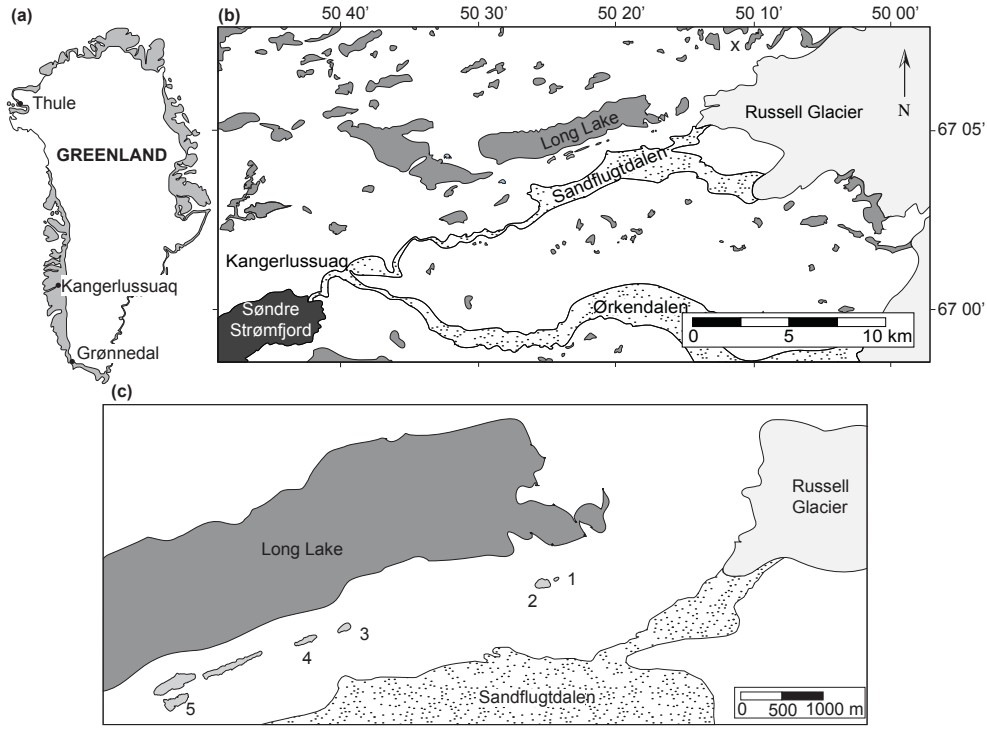

Figure 2. **a)** Greenland, showing Kangerlussuaq; **b)** Regional map of Kangerlussuaq and the inland ice margin, including Sandflugtdalen and Ørkendalen which are two major proglacial valley sandur systems. X marks the location of the Geological Survey of Denmark and Greenland (GEUS) weather station. **c)** Study area map of lakes relative to the Russell Glacier, Long Lake and Sandflugtdalen sandur. 1) EVV Upper lake; 2) EVV Lower lake; 3) Teardrop lake; 4) Potentilla lake; 5) South Twin lake





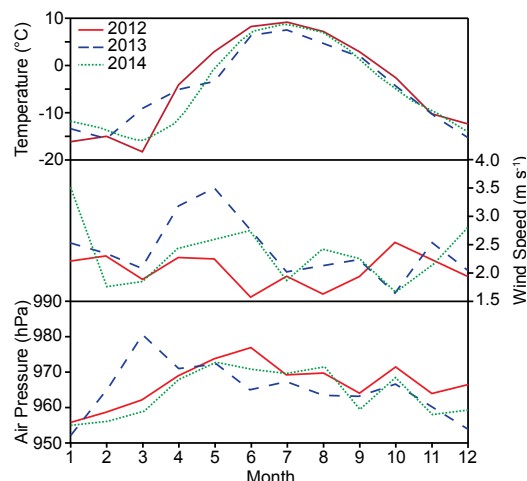

Figure 3. Mean monthly temperature, wind speed and air pressure for Kangerlussuaq from 2012 (red solid line), 2013 (blue dashed line) and 2014 (green dotted line). Data from the Program for Monitoring of the Greenland Ice Sheet (PROMICE) and the Greenland Analog Project (GAP) were provided by the Geological Survey of Denmark and Greenland at http://www.promice.dk.





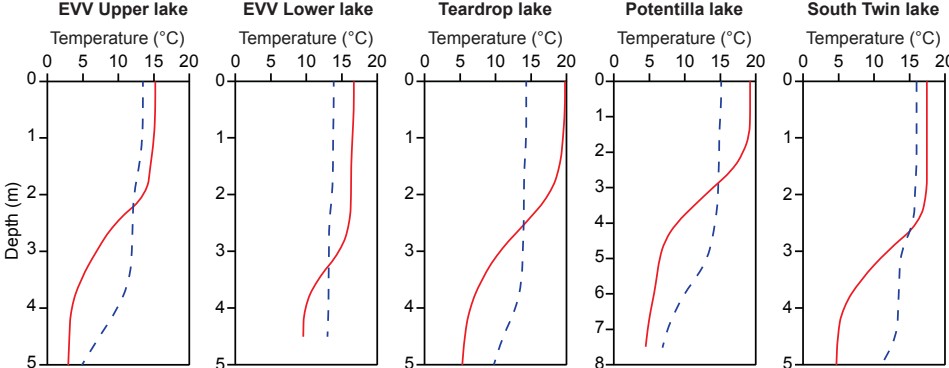


Figure 4. Profiles of temperature under open-water conditions in 2012 (red solid line) and 2013 (blue dashed line).





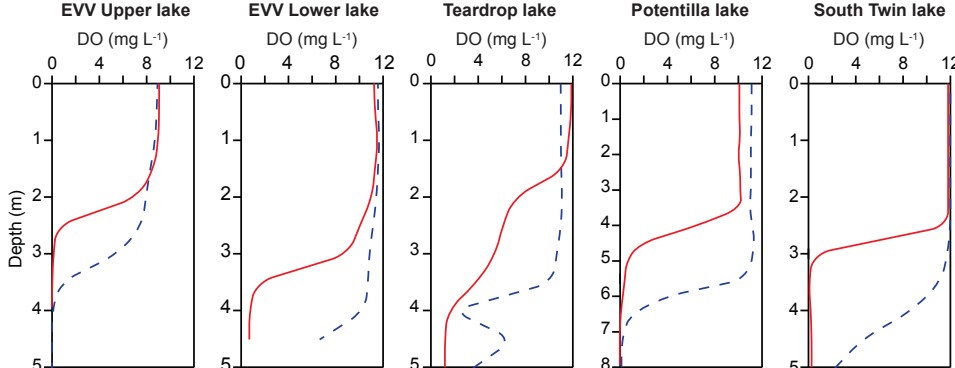

Figure 5. Profiles of DO under open-water conditions in 2012 (red solid line) and 2013 (blue dashed line).






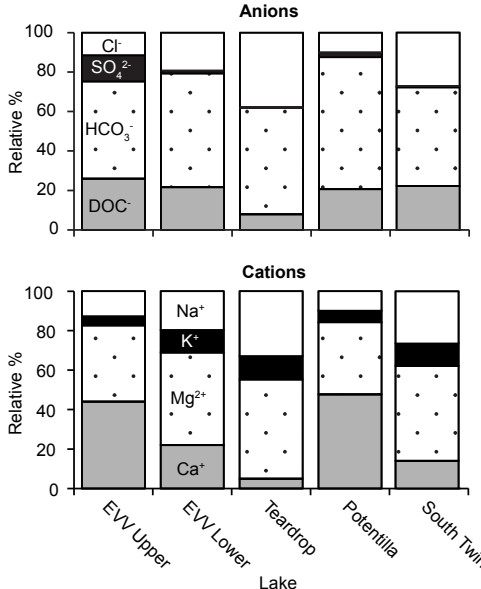

Figure 6. Relative percentages of median total charge associated with anions and cations in water samples from each of the study lakes. We assign the equivalents of missing anionic charge to DOC (Driscoll and Newton, 1985). In all study lakes, $HCO_3^-$ was the dominant anion. $SO_4^{2-}$ was the least abundant anion in all lakes except EVV Upper lake. There was no dominant cation from lake-to-lake.





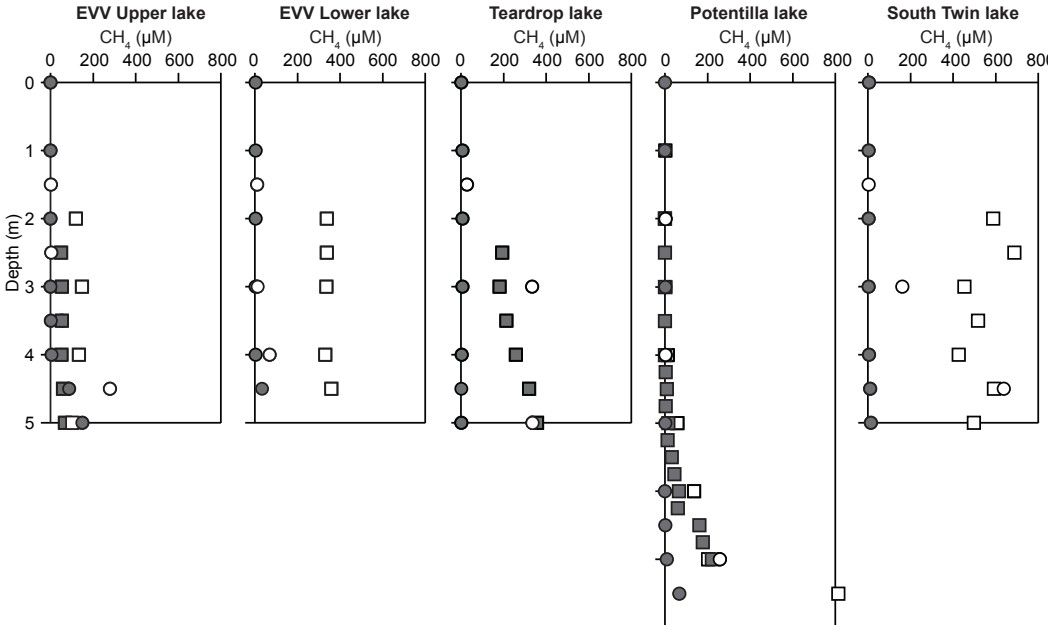

Figure 7. Profiles of dissolved $CH_4$ under open water conditions (circles) in July 2012 (open) and July 2013 (closed) and ice-covered conditions (squares) in April 2013 (open) and April 2014 (closed).






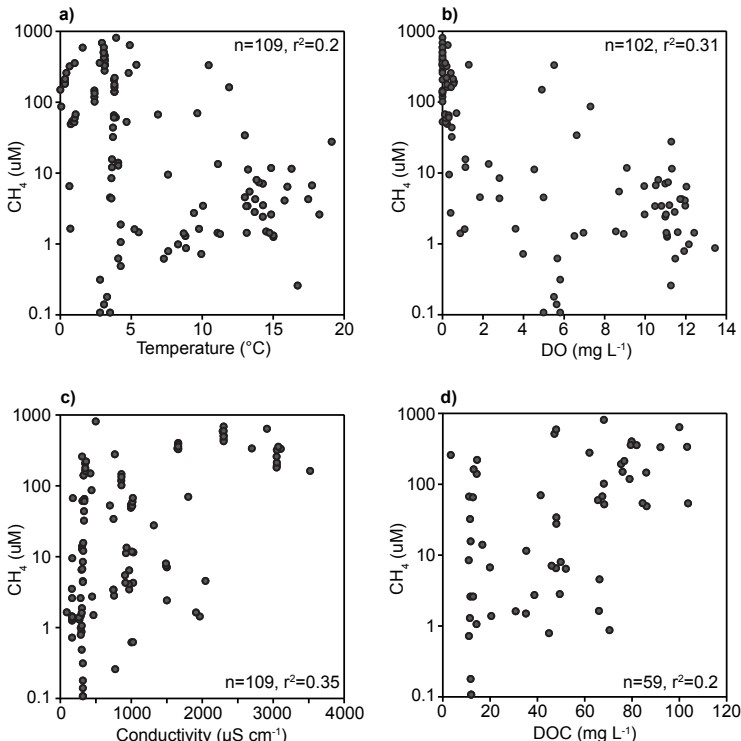

Figure 8. Relationships between dissolved $CH_4$ concentrations from 2012-2014 and **a)** temperature, **b)** DO, **c)** conductivity, and **d)** DOC.





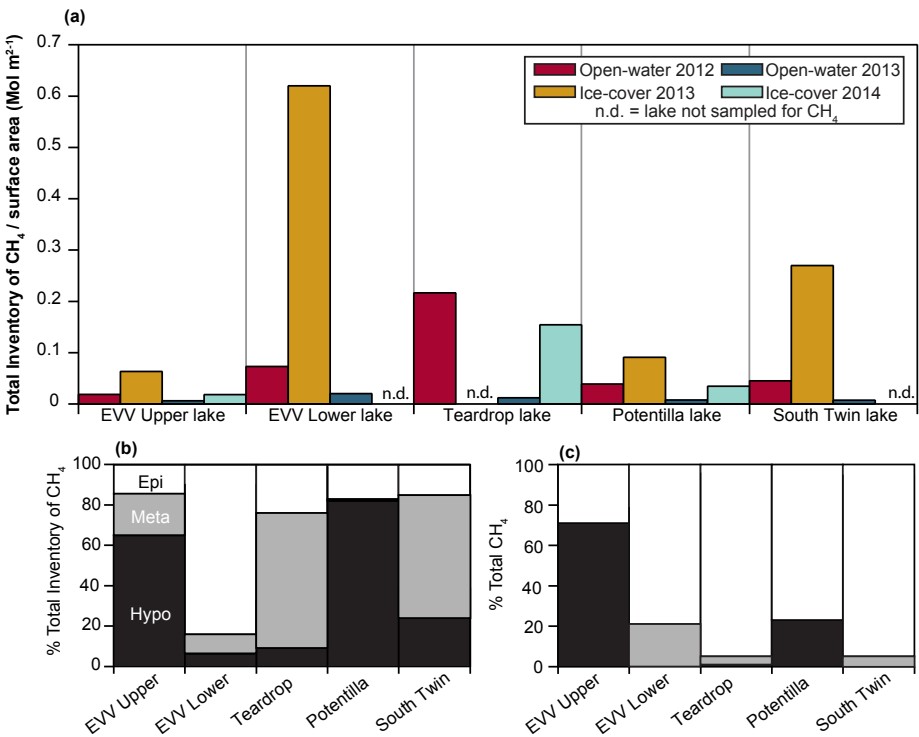


Figure 9. **a)** Total inventory (moles) of $CH_4$ in the water column of each lake under both open-water conditions in
July 2012 (black) and July 2013 (grey) and ice-covered conditions in April 2013 (white) and April 2014 (hashed-
lines). For seasons where the inventory was low, total moles are written above the bars; n.d. refers to seasons when a
given lake was not measured. **b & c) Relative** pool sizes (% of total) of dissolved $CH_4$ under open water conditions

in July 2012 (b) and July 2013 (c). Pools are defined by redox conditions, where the stored pool (black) is the sum
of $CH_4$ from intervals where DO < 1 mg L$^{-1}$, gray represents the suboxic pool, and the active pool (white) is the sum
of $CH_4$ from well mixed surface intervals potentially available for direct exchange with the atmosphere.





**11 Tables**


Table 1. Morphometrics and median physico-chemical characteristics of lakes under open-water conditions in 2012.

|  | $Z_{max}$ (m) | Fetch (m) | Surface area (ha) | Volume ($m^3$) | pH | Specific Conductivity ($mS\ cm^{-1}\ °C$) | DOC ($mg\ L^{-1}$) |
|---|---|---|---|---|---|---|---|
| EVV Upper | 5.5 | 68 | 0.22 | 5,200 | 7.0 | 0.7 | 38.7 |
| EVV Lower | 4.5 | 180 | 1.5 | 31,000 | 8.9 | 1.2 | 38.4 |
| Teardrop | 5.25 | 160 | 0.97 | 34,000 | 9.2 | 4.3 | 97.6 |
| Potentilla | 8.0 | 280 | 1.6 | 160,000 | 7.2 | 0.4 | 11.4 |
| South Twin | 5.5 | 310 | 3.1 | 120,000 | 7.9 | 4.0 | 20.5 |





Table 2. Seasonal and annual variation in median specific conductivity and DOC

| | Open Water | Ice Cover | Open Water | Ice Cover |
|---|---|---|---|---|
| | 2012 | 2013 | 2013 | 2014 |
| Specific Conductivity (mS cm$^{-1}$) | | | | |
| EVV Upper lake | 0.7 (10) | 1.5 (10) | 0.5 (11) | 1.0 (10) |
| EVV Lower lake | 1.2 (8) | 2.9 (7) | 1.0 (9) | n.d. |
| Teardrop lake | 4.3 (10) | n.d. | 1.9 (11) | 3.1 (10) |
| Potentilla lake | 0.4 (15) | 0.5 (15) | 0.3 (16) | 0.4 (13) |
| South Twin lake | 4.0 (10) | 4.0 (9) | 1.2 (9) | n.d. |
| DOC (mg L$^{-1}$) | | | | |
| EVV Upper lake | 39 (4) | 79 (3) | 48 (4) | 76 (6) |
| EVV Lower lake | 38 (2) | 80 (2) | 49 (2) | n.d. |
| Teardrop lake | 92 (3) | n.d. | 58 (4) | 77 (3) |
| Potentilla lake | 11 (3) | 17 (4) | 11 (3) | 12 (9) |
| South Twin lake | 21 (3) | 110 (2) | 48 (3) | n.d. |

n.d. refers to seasons when a given lake was not measured

sample size is in parentheses





Table 3. Ranges of CH$_4$ concentrations in profundal sediments and littoral sediment gas bubbles

|  | Profundal (mM) | | Littoral (ppmv) | |
| --- | --- | --- | --- | --- |
|  | Ice Cover | Open Water | Open Water | Open Water |
|  | 2013 | 2013 | 2012 | 2013 |
| EVV Upper lake | 0.7-1.9 (6) | 0.3-0.7 (5) | n.d. | 473,000 (1) |
| EVV Lower lake | 2.7 (1) | 0.8-1.9 (5) | n.d. | 332,000 (1) |
| Teardrop lake | n.d. | 0.5-1.2 (5) | 378,000 (1) | 372,000-404,000 (2) |
| Potentilla lake | 1.5-2.0 (4) | 1.0-1.4 (5) | 320,000 (1) | 166,000-262,000 (2) |
| South Twin lake | 1.4-3.4 (3) | 0.4-2.2 (4) | 434,000 (1) | 228,000-238,000 (2) |


n.d. refers to seasons when a given lake was not measured

sample size is in parentheses