# Peer review of "“Exceptional summer warming leads to contrasting outcomes for methane cycling in small Arctic lakes of Greenland”"

_Biogeosciences, 2016_

## Referee Comment (RC1) · Anonymous Referee #1 · 15 Aug 2016

Summary

The manuscript describes a study on dissolved CH4 concentration in five arctic shallow lakes located in Greenland. They used here five data sets (from summer 2012 to winter 2014) on the Southwest ice-free margin of Greenland. The aim of the study was focused on the effect of one high warming event occurred in summer 2012 on CH4 concentration profiles and compared it with subsequent years (2013 and 2014). The study of CH4 dynamics in lakes is a topic of broad scientific interest as lakes represent an important source of this gas to the atmosphere.

I recognize that it is a difficult task to study lakes in these extreme environments, and data coming from them are therefore valuable. The manuscript is not very clear in

demonstrating how the warm 2012 summer influenced CH4 dynamics in these lakes. Even, this study shows minor effects of the 2012 warm summer on CH4 dynamics (showed in Figure 7), and it is very difficult to correlate the minor effects to any particular phenomena (showed in Figure 8).

Likewise, the authors should always make clear when data have been previously published. I was surprised that several data in Tables, Figures (Figure 6 and 7) and Map (Figure 2) are the same (or at least very similar) than those reported in another manuscript from the same authors (Cadieux et al. 2016); and no reference is made to that previous study (and/or indicated in those tables and figures). I also want to point out, that there are strong similarities in the DOC and pH data presented in Table 1 and Table 2 (for DOC) for open-water conditions 2012 in this manuscript and data presented in Table 1 for open water conditions in 2013 from Cadieux et al. (2016).

The manuscript is well written, although some sections are not totally mature yet and therefore the manuscript lacks a clear focus and structure. I think that some of the analysis are speculative and/or over-interpreted and numerous issues in the method section must be better addressed.

Specific comments

The introduction contains mixed statements related to temperature effects on CH4 production/oxidation/storage in the water column (e.g. temperature dependencies on CH4 production is described in two sentences in second and fourth paragraphs). I would recommend reorganizing the ideas to improve the introduction flow (which should go from general to specific).

Likewise, it is necessary to carefully review the literature to avoid controversial statements like the authors indicate at the end of the introduction "This work provides the first measurements of dissolved CH4 concentrations under both open-water and ice-covered conditions for consecutive years in small, Arctic lakes". From the literature that I know (and for sure I am missing a vast amount of studies), there are previous studies

or multiyear dissolved CH4 concentrations in water column, in similar latitudes. Some of these previous works measured dissolved CH4 concentration through and over several years. I suggest some readings: Kaankala et al. (2006), Bellido et al. (2009), Karlsson et al. (2013), Greene et al. (2014), Miettinen et al. (2015), Tan et al. (2015), among others.

The description of the methods is the most important section to understand what the authors did. This section has to be improved substantially. Firstly, I found a number of cases in which devices or sample preparation are not full described (e.g. electronic submersible pump, total organic carbon analyzer, passive diffusion bags PDBs, HCl concentration, dilution correction for CH4 measurements). Secondly, littoral sediment CH4 bubble sampling method (used in this manuscript) is a very unspecific method. While in Cadieux et al. (2016) the method was used in combination with the isotopic analysis (isotopic values are helping to understand CH4 dynamics), in this manuscript, values of CH4 are given without determining the volume of sediment samples (as commented in the method section). Therefore, what is the point to include very speculative values of CH4 concentration from the littoral. Thirdly, I consider it would be necessary to describe briefly the methods, even if they are previously described (Cadieux et al. 2016), to avoid excessive self-citation and tedious reading. Finally, the statistical analyses need to be clarified. Some of them does not make sense, as written, and specific information is required to understand how data analysis was made e.g., mean/median temperature and CH4, profile values, seasonal, sectional.

Through the results and discussion section some Figures are used to explain variations and significant differences between lakes. In data from Figure 7, it is impossible to note the range reported in surface waters and depth axes are missing in some sub-figures (making impossible to see clearly the depth profile). Moreover, in data analysis from Figure 8 (wrongly named Figure 7 in Page 9, Line 307), it is impossible to see when CH4 vs. DO and CH4 vs. T are related or not. Likewise, some discussion sections are not well focused on the main issue and over interpret results. Some examples are:

i) "The competition for substrates favors sulfur reduction (SR) and methanogenesis typically does not occur until $SO_4^{2-}$ is exhausted and SR rates have decreased (Lovely & Klung 1983, Lovely & Klung 1986, Scholten et al., 2002, Ward & Winfrey 1985). However, EVV Upper lake did not have the lowest concentrations of $CH_4$ in the water column, suggesting there was sufficient reduced carbon substrates to fuel both SR and methanogenesis. Therefore, while aquatic chemistry in the water 320 column may be a factor influencing $CH_4$ production, it alone is insufficient to explain the variation in $CH_4$ concentrations observed lake-to-lake, as well as seasonally and annually."

ii) all section "6.3 Effects of temperature on $CH_4$", and

iii) you don't have thorough information on the ice phenology to indicate that "Our results suggest that changes in the duration of seasonal ice cover will, in turn, result in changes in inventories of under-ice $CH_4$. As the duration of ice cover decreases, the amount of $CH_4$ stored under ice 455 cover will likely decrease due to the shorter time for accumulation, potentially reducing the amounts of $CH_4$ emitted during ice-breakup and spring overturn.". I think, the results are not reliable to support such statements.

References

Bellido, J. L., Tulonen, T., Kankaala, P., and Ojala, A.: $CO_2$ and $CH_4$ fluxes during spring and autumn mixing periods in a boreal lake (Paajarvi, southern Finland), J. Geophys. Res.-Biogeosci., 114, G04007, doi:10.1029/2009JG000923, 2009.

Cadieux, S.B., White, J.R., Sauer, P.E., Peng, Y., Goldman, A.E. and Pratt, L.M.: Large fractionations of C and H isotopes related to methane oxidation in Arctic lakes, Geochim. Cosmochim. Ac., 187, 141-155, 2016.

Greene, S.,Walter Anthony, K. M., Archer, D., Sepulveda-Jauregui, A., and Martinez-Cruz, K.: Modeling the impediment of methane ebullition bubbles by seasonal lake ice, Biogeosciences, 11, 6791–6811, doi:10.5194/bg-11-6791-2014, 2014.

Kankaala, P., Huotari, J., Peltomaa, E., Saloranta, T., and Ojala, A.: Methanotrophic

activity in relation to methane efflux and total heterotrophic bacterial production in a stratified, humic, boreal lake, Limnol. Oceanogr., 51, 1195–1204, 2006.

Karlsson, J., Giesler, R., Persson, J., Lundin, E.: High emission of carbon dioxide and methane during ice thaw in high latitude lakes, J. Geophys. Res. Lett., 40, 1–5, doi:10.1002/grl.50152, 2013.

Miettinen, H., Pumpanen, J., Heiskanen, JJ., Aaltonen, H., Mammarella, I., Ojala, A., Levula, J., and Rantakari, M.: Towards a more comprehensive understanding of lacustrine greenhouse gas dynamics — two-year measurements of concentrations and fluxes of CO2, CH4 and N2O in a typical boreal lake surrounded by managed forests, Boreal Environ. Res. 20, 75–89, 2015.

Tan, Z., Zhuang, Q., Water Anthony, K.: Modeling methane emissions from arctic lakes: Model development and site-level study, J. Adv. Model. Earth Syst., 07, doi:10.1002/2014MS000344., 2015. (note: for sure to develop the models, they used multi-year dissolved CH4 concentration data).

---

## Referee Comment (RC2) · Anonymous Referee #2 · 19 Aug 2016

With a little focusing this interesting study could be a gem. The study demonstrates two mechanisms by which warming temperatures in Greenland could affect methane dynamics in small lakes. The first is during open water, increased stratification of the water column, which would presumably result in greater methane release during fall overturn and less overall methane oxidation. The second mechanism is that greater temps will result in less overall ice cover which would cause less methane storage under the ice and presumably more overall oxidation. These two processes or effects of increased temperature would seemingly have contradictory effects. I don't know the extent to which these two processes effects have been expounded in the literature, but this is the first time I've seen them presented. I would thus suggest to the

authors that they make more of these observations, highlighting them in the abstract, and particularly in the article titles, which is rather weak right now, in my opinion. Perhaps something like " The contradictory nature of warming effects on lake methane emissions: increased stratification during ice free periods versus reduced ice cover." Needs work, but something along those lines. I would also suggest that these unique observations be expanded into a conceptual model in the discussion and conclusions.

Specific comments.

1. abstract. See above. ALSO focus on the effects these processes will have on overall annual lake methane emissions. That's what's important. You may not have the data, but speculate, and call for attention to what your have observed so it can be followed up.

2. page 3, lines 90-95. Your hypothesis. Why did you hypothesize that warmer conditions would lead to higher methane concentrations? Say "increased stratification" here. Explicitly state it. Advance a hypothesis about ice cover. Return to these hypotheses in your discussion.

3. Lines 95-100. Permafrost soil? Anything you can tell us about it? Peat? Mineral soil? OM content? Does it thaw under the lake (thaw bulb) to make the methane you observe?

4. line 112. define GIS

5. line 126 define EVV

6. line 211. what is Clinograde?

7. line 225. "moderately brackish salinity? What was the salinity in o/oo? Is "brackish" the right term? Like 5-10 o/oo?

8. Line 307-308. confusing. Is the sentence messed up?

9. Line 308 do you mean figure 8? Not 7?

10. Line 315 sulfate reduction not sulfur.

11. Line 352. Inversely related?

12. Lines 415-425. I don't follow this too well. How do you know that the % of CH4 oxidized is the same over the two years? Do you have measurements of MOX> ? Doesn't this kind of blow your theory that more temp and more stratification will result in more methane release with fall overturn? How is the MOX the same across years, or even known at all?

13. Develop around line 425-460 the effects of a shorter ice covered period. Make a solid conceptual model centered on your figures. What is the interplay between increased stratification during ice free in contrast with less ice cover? How does this interplay affect annual methane flux as temperatures warm> I would think that there would be less MOX under stratified conditions, and certainly less under ice.

14. Conclusions. Point out that these two processes are contradictory.

---

## Author Comment (AC1) · 18 Oct 2016

*response from authors is in blue

**Summary**
The manuscript describes a study on dissolved CH4 concentration in five arctic shallow lakes located in Greenland. They used here five data sets (from summer 2012 to winter 2014) on the Southwest ice-free margin of Greenland. The aim of the study was focused on the effect of one high warming event occurred in summer 2012 on CH4 concentration profiles and compared it with subsequent years (2013 and 2014). The study of CH4 dynamics in lakes is a topic of broad scientific interest as lakes represent an important source of this gas to the atmosphere.

I recognize that it is a difficult task to study lakes in these extreme environments, and data coming from them are therefore valuable. The manuscript is not very clear in demonstrating how the warm 2012 summer influenced CH4 dynamics in these lakes. Even, this study shows minor effects of the 2012 warm summer on CH4 dynamics (showed in Figure 7), and it is very difficult to correlate the minor effects to any partic- ular phenomena (showed in Figure 8).

Likewise, the authors should always make clear when data have been previously published. I was surprised that several data in Tables, Figures (Figure 6 and 7) and Map (Figure 2) are the same (or at least very similar) than those reported in another manuscript from the same authors (Cadieux et al. 2016); and no reference is made to that previous study (and/or indicated in those tables and figures). I also want to point out, that there are strong similarities in the DOC and pH data presented in Table 1 and Table 2 (for DOC) for open-water conditions 2012 in this manuscript and data presented in Table 1 for open water conditions in 2013 from Cadieux et al. (2016).

The manuscript is well written, although some sections are not totally mature yet and therefore the manuscript lacks a clear focus and structure. I think that some of the analysis are speculative and/or over-interpreted and numerous issues in the method section must be better addressed.

First, we thank the reviewer for their positive comments on the paper and feel they have raised some valid concerns. The reviewer is correct in noticing that some data and parts of figures are in Cadieux et al. 2016. However, its important to note that in Cadieux et al. 2016, only 3 lakes (EVV Upper lake, Teardrop lake, and Potentilla lake) are discussed for only open water conditions in 2013 and ice covered conditions 2014. In order to be explicit about data that is previously published, we have revised tables and figures to cite which lakes and data points have been previously published.

The reviewer is also correct in noticing that in Table 1, these physico-chemical characteristics of the lakes in open-water conditions in 2012 are the same as in Cadieux et al. (2016) - which is inaccurately labeled water conditions for 2013 when they are 2012. For the paper here, we revised the table text to include that this data is also in Cadieux et al. 2016 and will take steps to revise the other paper so both match.

**Specific comments**

The introduction contains mixed statements related to temperature effects on CH4 production/oxidation/storage in the water column (e.g. temperature dependencies on CH4 production is described in two sentences in second and fourth paragraphs). I would recommend reorganizing the ideas to improve the introduction flow (which should go from general to specific).
Thank you for this suggestion to strengthen the introduction. In order to go from general to specific, as well as remove repetitive information, the organization now goes from discussion of lakes and climate, to methane production and consumption, to focus of this study.

Likewise, it is necessary to carefully review the literature to avoid controversial state- ments like the authors indicate at the end of the introduction "This work provides the first measurements of dissolved CH4 concentrations under both open-water and ice- covered conditions for consecutive years in small, Arctic lakes". From the literature that I know (and for sure I am missing a vast amount of studies), there are previous studies or multiyear dissolved CH4 concentrations in water column, in similar latitudes. Some of these previous works measured dissolved CH4 concentration through and over sev- eral years. I suggest some readings: Kaankala et al. (2006), Bellido et al. (2009), Karlsson et al. (2013), Greene et al. (2014), Miettinen et al. (2015), Tan et al. (2015), among others.

We appreciate the reviewer bringing this to our attention. Previously, in our literature search, we had not found Karlsson et al. (2013), Miettienen et al. 2015 and Greene et al. 2014 which all also describe both multiyear and multiseason results. The others described above are only for one season or one year. We have removed this sentence accordingly.

The description of the methods is the most important section to understand what the authors did. This section has to be improved substantially. Firstly, I found a number of cases in which devices or sample preparation are not full described (e.g. electronic submersible pump, total organic carbon analyzer, passive diffusion bags PDBs, HCl concentration, dilution correction for CH4 measurements). Secondly, littoral sediment CH4 bubble sampling method (used in this manuscript) is a very unspecific method. While in Cadieux et al. (2016) the method was used in combination with the isotopic analysis (isotopic values are helping to understand CH4 dynamics), in this manuscript, values of CH4 are given without determining the volume of sediment samples (as com- mented in the method section). Therefore, what is the point to include very speculative values of CH4 concentration from the littoral. Thirdly, I consider it would be necessary to

describe briefly the methods, even if they are previously described (Cadieux et al. 2016), to avoid excessive self-citation and tedious reading. Finally, the statistical anal- yses need to be clarified. Some of them does not make sense, as written, and specific information is required to understand how data analysis was made e.g., mean/median temperature and CH4, profile values, seasonal, sectional.

The reviewer makes good point that this paper would be strengthened by describing the methods in more detail. We revised the methods section in the following ways in an attempt to address the concerns noted above:

- Regarding the DOC measurement, a reference has been added to describe the questions of methodology (Oviedo-Vargas, D., Royer, T.V., Johnson, L.T., 2013. Dissolved organic carbon manipulation reveals coupled cycling of carbon, nitrogen, and phosphorus in a nitrogen-rich stream. Limnology and Oceanography 58, 1196-1206.).

- We specified the model and type of electronic submersible pump: "*Water for chemical analysis was collected from the water column using a Narrow Diameter Supernova™ electronic submersible pump.*"

- The methods for dissolved methane sampling have been briefly expanded: "*With the exception of Potentilla lake under ice-covered conditions in 2014, water samples for dissolved CH$_4$ in the water column were collected using an electronic submersible pump. Samples were collected at 0.25-1.0 m intervals through the water column and were immediately stripped in the field using a headspace-equilibrium technique (Westendorp 1985) to extract CH$_4$ from water. At each depth interval, 500 mL of water was collected into a 1 L Erlenmeyer flask and vigorously shaken for 1 minute. Headspace gas in the flask was displaced into a Cali-5-Bond bag using surficial lake water (Cadieux et al., 2016). Under ice-covered conditions in 2014, dissolved CH$_4$ in Potentilla lake was collected using a string of passive diffusion bags (PDBs) deployed in the lake for 5 days in order to obtain a high-resolution profile of dissolved CH$_4$ in the water column (Goldman et al., 2016). The PDBs are composed of a polyethylene membrane with a protective plastic mesh and are commercially available from EON Products Inc. (Georgia, USA). After 5 days, PDBs were retrieved from the lake and dissolved gas was sampled immediately in the field using the equilibrium gas stripping method described above. Further details regarding PDB methodology, preparation and applicability can be found in Goldman et al. (2016).*"

- We acknowledge that the littoral methane concentrations are speculative, as we did not measure a concentration of sediment in order to normalize lake-to-lake. We have added a statement to explicitly state that this is just an estimation: "*We were unable to quantify the volume of sediment samples, therefore concentrations of CH$_4$ in gas collected from littoral sediments cannot be*

*converted into pool size of CH₄ in the littoral sediments, and are only an approximation of CH₄ concentration."*

- In order to not over-analyzed results, the following was added to the littoral CH₄ section: *"However, it is important to note that littoral CH₄ concentrations are an estimate, as a volume of sediment/sample was unmeasured. Therefore, it is possible that the increase in littoral CH₄ concentrations is not the result of increased CH₄ production, but of a different amount of sediment disturbed."*

- Statistical analysis section has been clarified as to what was tested and why: *"Statistical analyses were made using IBM SPSS Statistics. Concentrations of CH₄ and chemical variables for all study lakes during each season were assessed for normal distribution via the Kolmogorov-Smirnov test, and were found to be non-normally distributed. Student's t test of unequal variance was used for testing statistically significant differences in concentrations of CH₄ between open-water and ice-covered conditions, as well as from one year to another. Systematic changes in aquatic chemistry and CH₄ concentrations were analyzed using linear regression, in order to assess if CH₄ concentrations were related to variables such as DO, temperature, DOC and conductivity."*

Through the results and discussion section some Figures are used to explain variations and significant differences between lakes. In data from Figure 7, it is impossible to note the range reported in surface waters and depth axes are missing in some sub-figures (making impossible to see clearly the depth profile). Moreover, in data analysis from Figure 8 (wrongly named Figure 7 in Page 9, Line 307), it is impossible to see when CH4 vs. DO and CH4 vs. T are related or not. Likewise, some discussion sections are not well focused on the main issue and over interpret results. Some examples are:

Regarding the figures, Line 307 was revised to Figure 8 and depth axes have been added to all of the panels in figure 7.

We agree with the reviewer that it is difficult to note the range of surface water CH₄ concentrations in Figure 7. In order to clarify this, a new table has been made that defines the surficial values under open-water conditions in all of the lakes:

Table 3: CH₄ concentrations (µM) in surface waters under open-water conditions in 2012 and 2013.

|            | 2012 | 2013 |
|------------|------|------|
| EVV Upper  | 2.7  | 0.9  |
| EVV Lower  | 11.5 | 2.7  |
| Teardrop   | 27.8 | 2.4  |
| Potentilla | 2.6  | 1.3  |
| South Twin | 4.3  | 3.5  |

We also agree with the reviewer that in Figure 8 it is difficult to see the trend. This is because there is not a very statistically significant trend between CH₄ concentration and

temperature/dissolved oxygen. This is why we had included $r^2$ values on each of the figures. We have gone through the text to ensure that we do not over analyze this non-significant trend.

i) competition for substrates favors sulfur reduction (SR) and methanogenesis typically does not occur until SO42- is exhausted and SR rates have decreased (Lovely & Klung 1983, Lovely & Klung 1986, Scholten et al., 2002, Ward & Winfrey 1985). However, EVV Upper lake did not have the lowest concentrations of CH4 in the water column, suggesting there was sufficient reduced carbon substrates to fuel both SR and methanogenesis. Therefore, while aquatic chemistry in the water 320 column may be a factor influencing CH4 production, it alone is insufficient to explain the variation in CH4 concentrations observed lake-to-lake, as well as seasonally and annually."

Give the small sample size for each lake, our statistical power is limited for aquatic chemistry. In keeping with the reviewer's suggestion, the last sentence has been revised to: "*Therefore, while aquatic chemistry in the water column could be a factor influencing $CH_4$ production, at the level of this investigation, it alone is likely insufficient to explain the variation in $CH_4$ concentrations observed lake-to-lake, as well as seasonally and annually.*"

ii) all section "6.3 Effects of temperature on CH4", and

In keeping with the reviewer's suggestion, we have revised section 6.3 to explicitly mention that the relationships observed are at a specific date and time. For example: "*Despite the absence of a strong linear relationship between water temperature and $CH_4$ concentrations, warmer ground-level air temperatures correspond with increased $CH_4$ both in the water column and the sediments in the study lakes at the time of sampling.*"

iii) you don't have thorough information on the ice phenology to indicate that "Our re- sults suggest that changes in the duration of seasonal ice cover will, in turn, result in changes in inventories of under-ice CH4. As the duration of ice cover decreases, the amount of CH4 stored under ice 455 cover will likely decrease due to the shorter time for accumulation, potentially reducing the amounts of CH4 emitted during ice-breakup and spring overturn.". I think, the results are not reliable to support such statements.

We agree with the reviewer that we don't have dates for ice-in or ice-out or other ice phenology information. However, with the data we do have, we can speculate what may happen to $CH_4$ inventories as ice-cover duration decreased. In keeping with the reviewer's comments, we revised this section accordingly: "*Currently, the largest efflux of $CH_4$ from our study lakes occurs during ice-breakup and spring overturn. Changes in the duration of seasonal ice-cover will result in changes in inventories of under-ice $CH_4$. We predict that as the duration of ice cover decreases, the amount of $CH_4$ stored under ice cover will likely decrease due to the shorter time for accumulation.  If the amount of stored $CH_4$ under ice-cover decreases, this will potentially reduce the amount of $CH_4$ emitted during ice-breakup and spring overturn.*"

References

Bellido, J. L., Tulonen, T., Kankaala, P., and Ojala, A.: CO2 and CH4 fluxes during spring and autumn mixing periods in a boreal lake (Paajarvi, southern Finland), J. Geophys. Res.-Biogeosci., 114, G04007, doi:10.1029/2009JG000923, 2009.

Cadieux, S.B., White, J.R., Sauer, P.E., Peng, Y., Goldman, A.E. and Pratt, L.M.: Large fractionations of C and H isotopes related to methane oxidation in Arctic lakes, Geochim. Cosmochim. Ac., 187, 141-155, 2016.

Greene, S.,Walter Anthony, K. M., Archer, D., Sepulveda-Jauregui, A., and Martinez- Cruz, K.: Modeling the impediment of methane ebullition bubbles by seasonal lake ice, Biogeosciences, 11, 6791–6811, doi:10.5194/bg-11-6791-2014, 2014.

Kankaala, P., Huotari, J., Peltomaa, E., Saloranta, T., and Ojala, A.: Methanotrophic activity in relation to methane efflux and total heterotrophic bacterial production in a stratified, humic, boreal lake, Limnol. Oceanogr., 51, 1195–1204, 2006.

Karlsson, J., Giesler, R., Persson, J., Lundin, E.: High emission of carbon dioxide and methane during ice thaw in high latitude lakes, J. Geophys. Res. Lett., 40, 1–5, doi:10.1002/grl.50152, 2013.

Miettinen, H., Pumpanen, J., Heiskanen, JJ., Aaltonen, H., Mammarella, I., Ojala, A., Levula, J., and Rantakari, M.: Towards a more comprehensive understanding of lacustrine greenhouse gas dynamics two-year measurements of concentrations and fluxes of CO2, CH4 and N2O in a typical boreal lake surrounded by managed forests, Boreal Environ. Res. 20, 75–89, 2015.

Tan, Z., Zhuang, Q., Water Anthony, K.: Modeling methane emissions from arc- tic lakes: Model development and site-level study, J. Adv. Model. Earth Syst., 07, doi:10.1002/2014MS000344., 2015. (note: for sure to develop the models, they used multi-year dissolved CH4 concentration data).

We thank the reviewer for including the full citations for the references suggested. These have been incorporated into the text.

---

## Author Comment (AC2) · 18 Oct 2016

*response from authors is in blue

With a little focusing this interesting study could be a gem. The study demonstrates two mechanisms by which warming temperatures in Greenland could affect methane dynamics in small lakes. The first is during open water, increased stratification of the water column, which would presumably result in greater methane release during fall overturn and less overall methane oxidation. The second mechanism is that greater temps will result in less overall ice cover which would cause less methane storage un- der the ice and presumably more overall oxidation. These two processes or effects of increased temperature would seemingly have contradictory effects. I don't know the extent to which these two processes effects have been expounded in the literature, but this is the first time I've seen them presented. I would thus suggest to the authors that they make more of these observations, highlighting them in the abstract, and particularly in the article titles, which is rather weak right now, in my opinion. Per-haps something like " The contradictory nature of warming effects on lake methane emissions: increased stratification during ice free periods versus reduced ice cover." Needs work, but something along those lines. I would also suggest that these unique observations be expanded into a conceptual model in the discussion and conclusions.

Thank you for this positive review of our paper.

The reviewer has a valid point that the title could be modified to be more compelling. We have taken this into consideration and revised the title to: *"Exceptional summer warming leads to contrasting outcomes for methane cycling in small Arctic lakes of Greenland"*

Specific comments.

1.  abstract. See above. ALSO focus on the effects these processes will have on overall annual lake methane emissions. That's what's important. You may not have the data, but speculate, and call for attention to what your have observed so it can be followed up.

We agree with the reviewer that adding more regarding the possible overall methane emissions and highlighting the shift from spring emission to fall emission is important to highlight in the abstract. The later part of the abstract has been revises as follows to include these suggestions: *"In all of the lakes, mean methane concentrations under ice-covered conditions were significantly (p < 0.0001) greater than under open-water conditions, suggesting spring overturn is currently the largest annual methane flux to the atmosphere. As the climate continues to warm, shorter ice cover durations are expected, which may reduce the winter inventory of methane and lead to a decrease in total methane flux during ice-melt. Under open-water*

*conditions, greater heat income and warming of lake surface waters will lead to increased thermal stratification and hypolimnetic anoxia, which will consequently result in increased water column inventories of methane. This stored methane will be susceptible to emissions during fall overturn, which may result in a shift in greatest annual efflux of methane from spring melt to fall overturn. The results of this study suggest that inter-annual variation in ground-level air temperatures may be the primary driver of changes in methane dynamics because it controls both the duration of ice over and strength of thermal stratification."*

2.  page 3, lines 90-95. Your hypothesis. Why did you hypothesize that warmer conditions would lead to higher methane concentrations? Say "increased stratification" here. Explicitly state it. Advance a hypothesis about ice cover. Return to these hypotheses in your discussion.

We hypothesized that warmer conditions would lead to higher methane concentrations, because warmer conditions would lead to increased stratification. This sentence has been revised to increased stratification in order to clarify.

The reviewer makes a good point that a hypothesis should be stated regarding ice cover. Accordingly, the following hypothesis has been added about ice cover: "*The study lakes are ice-covered for 9-10 months of the year, leading us to predict that methane concentrations would be significantly greater under ice-covered conditions as opposed to open-water conditions."*

3.  Lines 95-100. Permafrost soil? Anything you can tell us about it? Peat? Mineral soil? OM content? Does it thaw under the lake (thaw bulb) to make the methane you observe?

The soil is composed predominantly of till and glaciofluvial deposits. While there are talics in the area, our observations do not suggest there are talics below any of the study lakes here. In response to this comment, we have added the following sentence to further define soils in the region: "*Soils in the region are not well-developed, composed of till and glaciofluvial deposits (Van Tatenhove and Olesen, 1994)."*

4.  line 112. define GIS

GIS stands for the Greenland Ice Sheet, and has been defined in the text accordingly. It is also defined in the introduction.

5.  line 126 define EVV

EVV stands for Epidode Vein Valley and is an informal name to describe an outcrop close in proximity to the lakes. We have revised the study area section to include further information regarding the names of the lakes: "*The lake names used herein (EVV Upper Lake, EVV Lower Lake, Teardrop Lake, Potentilla Lake and South Twin Lake) are informal based on local surficial features"*

6. line 211. what is Clinograde?

A clinograde oxygen profile is when dissolved oxygen values decrease with depth. This term is defined in the later part of the sentence: *"wherein DO was saturated and in equilibrium with the atmosphere in the surface waters and became increasingly under saturated down the water column."*

7. line 225. "moderately brackish salinity? What was the salinity in o/oo? Is "brackish" the right term? Like 5-10 o/oo?

These are terms from Stewart and Kantrud paper, which has been added to the references. In terms of salinity, brackish here would be >2 ‰.

8. Line 307-308. confusing. Is the sentence messed up?

This sentence did need to be clarified. It has been revised to state: "*In the water column, $CH_4$ concentrations are directly related to both conductivity and DOC, wherein high $CH_4$ concentrations correspond with both high conductivity and DOC.*"

9. Line 308 do you mean figure 8? Not 7?

Yes, thank you

10. Line 315 sulfate reduction not sulfur.

Done. Thank you

11. Line 352. Inversely related?
Yes, this should be inversely related and has been revised.

12. Lines 415-425. I don't follow this too well. How do you know that the % of CH4 oxidized is the same over the two years? Do you have measurements of MOX> ? Doesn't this kind of blow your theory that more temp and more stratification will result in more methane release with fall overturn? How is the MOX the same across years, or even known at all?

The way we determined that the same amount of $CH_4$ was oxidized was by assuming all the methane in the surface water originated from the sediment. By doing that, what is at the surface is a percentage of the initial methane (at the sediment-water interface). Therefore, we don't have the measurements of MOX, just an estimate of the % of methane oxidized. We have revised this paragraph to emphasize that this isn't an estimate of MOX, but an estimate of the amount of methane oxidized.

Even though the amount oxidized at the time of stratification is the same, it doesn't change that under warmer conditions with increased stratification in 2012 there is more methane in

the anoxic waters, which is what would be emitted during fall overturn.

13. Develop around line 425-460 the effects of a shorter ice covered period. Make a solid conceptual model centered on your figures. What is the interplay between increased stratification during ice free in contrast with less ice cover? How does this interplay affect annual methane flux as temperatures warm> I would think that there would be less MOX under stratified conditions, and certainly less under ice.

We agree with the reviewers concern and expanded the relevant discussion to better articulate the conceptual model as informed by our results: "*In addition to a decrease in ice cover, our results also suggest an increase in ground-level air temperatures will result in enhanced thermal stability and anoxia in Arctic lakes, as we observed during open-water conditions in 2012. The duration of open-water thermal stratification will also likely increase in concert with the decrease in ice cover. The combined effects of extended season and greater strength of stratification are likely to be development of higher $CH_4$ inventories in the water column during open-water periods. Conceptually, as anoxic zones expand in space and duration, the influence of methanogenic sediments on water column inventories of methane should increase. Currently, small lakes emit substantially more $CH_4$ per unit area than larger lakes during open-water conditions (Bastviken et al., 2004; Cole et al., 2007; Juutinen et al., 2009). Small, shallow likes are more susceptible to thermal change due to increased ground-level air temperatures and will likely continue to be major $CH_4$ contributors to the atmosphere. In fact, our results suggest that increased warming in the Arctic will result in greater summer inventories of $CH_4$ and consequently larger emissions of $CH_4$ to the atmosphere during autumn overturn in small lakes.*"

14. Conclusions. Point out that these two processes are contradictory.

In response to this suggestion, the conclusions have been revised to explicitly point out that these two processes are contradictory, leading to the inclusion of the following sentence: "*We predict that as the climate continues to warm, the greatest annual efflux of $CH_4$ from small arctic lakes will shift from spring overturn to fall overturn.*"